# Belimumab after B cell depletion therapy in patients with systemic lupus erythematosus (BEAT Lupus) protocol: a prospective multicentre, double-blind, randomised, placebo-controlled, 52-week phase II clinical trial

Alexis Jones,[1] Patrick Muller [ID],[2] Caroline J Dore,[2] Felicia Ikeji,[2] Emilia Caverly,[2] Kashfia Chowdhury,[2] David A Isenberg [ID],[1] Caroline Gordon,[3] Michael R Ehrenstein [ID] [1]

For numbered affiliations see end of article.

**Correspondence to**
Professor Michael R Ehrenstein;
m.ehrenstein@ucl.ac.uk

## ABSTRACT

**Introduction** Few treatment options exist for patients with systemic lupus erythematosus (SLE) who fail conventional therapy. Although widely used to treat lupus, the efficacy of B cell depletion therapy using rituximab has not been demonstrated in randomised clinical trials. Following rituximab, elevated levels of serum B cell activating factor (BAFF) have been associated with failure to remit or subsequent lupus relapse. The administration of belimumab, a monoclonal antibody specific for BAFF and approved for lupus therapy, could potentiate the efficacy of rituximab and enable longer periods of disease remission. The aim of this trial is to assess the safety and efficacy of belimumab following rituximab in patients with SLE.

**Methods and analysis** BEAT Lupus is a double-blind, randomised, placebo controlled, phase II clinical trial. Patients with SLE commencing a treatment cycle of rituximab (two 1g infusions, 2 weeks apart) as standard of care will be randomised to receive belimumab or placebo, 4 to 8 weeks following the first rituximab infusion. Belimumab or placebo infusions are administered for 52 weeks. The primary outcome measure is anti-double stranded DNA (anti-dsDNA) antibody levels at 52 weeks. Secondary outcomes include measures of adverse events, lupus disease activity and cumulative steroid dose. The kinetics of B cell repopulation will be assessed in a subgroup of participants. Belimumab administration after rituximab may provide a novel therapeutic pathway for patients with active lupus if safety is demonstrated in this proof of concept study, and lower anti-dsDNA antibodies levels are achieved in those patients treated with belimumab compared with placebo.

**Ethics and dissemination** The protocol has been reviewed and approved by the Hampstead Research Ethics Committee - London (reference 16/LO/1024). Trial information is available at https://www.isrctn.com/ISRCTN47873003, and the results of this trial will be submitted for publication in relevant peer-reviewed journals. Key findings will also be presented at national and international conferences.

### Strengths and limitations of this study

► Double-blind, randomised, placebo controlled trial in lupus patients refractory to conventional therapy thereby reflecting real world practise while eliminating expectation of treatment benefit.
► Trial recruitment at point of randomisation to belimumab or placebo, rather than before rituximab, to improve safety and mitigate loss of patients due to adverse events following rituximab.
► Small sample size because of safety considerations prevailing at the conception of the trial may limit analyses.
► Trial design stipulates reduction in concomitant steroid and immunosuppressant dosage to maximise any potential difference between belimumab and placebo.
► The experimental medicine component of this trial will enable a greater understanding of response to combination B cell depletion and belimumab therapy.

**Trial registration number** ISRCTN47873; date assigned to the registry: 28 November 2016. The stage is pre-results.

## INTRODUCTION

Despite burgeoning research into the pathogenesis of systemic lupus erythematosus (SLE), the introduction of novel treatments into the clinic has been slow. This is in contrast to some other immune mediated rheumatic diseases, in particular rheumatoid arthritis (RA), for which several novel therapies developed through increased understanding of disease pathogenesis have made a substantial impact on patient care. A number

of reasons may account for the disparity between SLE and RA including the greater use of corticosteroids which limits the potential for novel treatments to show superiority in clinical trials; the multisystem nature of lupus and insufficient understanding of the key factors that drive disease.[1]

Rituximab, a chimeric anti-CD20 monoclonal antibody, has shown significant, though variable, efficacy in open-label studies, including both single-centre[2–5] and multi-centre studies[6–9]; as well as a systematic review of off-label use.[6] However, two large, phase III, randomised placebo-controlled trials in non-renal lupus (EXPLORER)[10 11] and renal lupus (LUNAR),[12] failed to meet their primary endpoints. Nevertheless, the European League Against Rheumatism have accepted that therapy with rituximab should be considered in organ-threatening, refractory lupus.[13] In addition, National Health Service (NHS) England have approved rituximab as part of standard of care in patients with moderate-to-severe lupus who have failed conventional immunosuppressant therapy due to the few treatment options available for this patient group (NHS ENGLAND A13/PS/a).[14] However, the variable therapeutic success of rituximab highlights the need to refine therapies that target B cells to improve the outcome for patients with SLE.

The efficiency of B cell depletion following rituximab treatment has been shown to predict response in patients with lupus.[4 15 16] Analysis of peripheral B cell subsets revealed that patients with lower plasmablast counts at 6 months are more likely to have a sustained response without requiring retreatment.[4] In the subset of SLE patients with high levels of anti-double stranded DNA (anti-dsDNA) antibodies, clinical relapse occurs at lower B cell numbers after treatment compared with before rituximab and is associated with higher proportions of repopulating plasmablasts in the peripheral blood.[15] Further investigation into the mechanism of rituximab has demonstrated that levels of the B cell cytokine BAFF (B cell activating factor) surge after B cell depletion.[17 18] Moreover, elevated serum BAFF levels can endure beyond initial B cell repopulation and distinguish lupus relapse from ongoing disease remission following rituximab.[19] In some lupus patients repeated cycles of rituximab resulted in ever higher levels of serum anti-dsDNA antibodies, which were associated with increasing levels of serum BAFF.[16] This observation raises the possibility that repeated rituximab treatments may result in more severe flares driven by BAFF.[20] Thus, high BAFF levels post rituximab could be limiting its effectiveness in some patients with SLE.

Clinical studies have demonstrated that BAFF is over-expressed, either intermittently or persistently, in many patients with SLE irrespective of rituximab therapy[21 22] and serum levels correlate with serological and clinical indices of disease activity.[23] In 2016, the BAFF-neutralising monoclonal antibody, belimumab was the first biological licensed for the treatment of lupus following two large phase III clinical trials, BLISS 52[24] and BLISS 76.[25] Treatment with belimumab after B cell depletion has been suggested as a potential therapeutic strategy for patients with SLE.[20] We therefore designed a phase II clinical trial to assess the safety and effectiveness of this treatment regime for patients with lupus, the latter judged by a change in anti-dsDNA antibody levels as the primary outcome. Anti-dsDNA antibodies have been associated with disease activity in patients with SLE.[14 26 27] Secondary outcomes include incidence of adverse events, reduction in cumulative steroid intake and incidence of disease flares during the 52 week follow-up period. This trial will also assess the kinetics of B cell repopulation over the course of the trial.

## METHODS AND ANALYSIS

BEAT Lupus is a phase IIb, multicentre, UK based, randomised, double blind, placebo-controlled interventional clinical trial investigating the safety and efficacy of belimumab (given for 12 months), 4 to 8 weeks after the first infusion of B cell depletion therapy (rituximab). Rituximab is administered according to the NHS England interim clinical commissioning policy (NHS England A13/PS/a) and the British Society for Rheumatology guideline for the management of systemic lupus erythematosus in adults,[14] specifically two 1 g doses of rituximab given intravenously 2 weeks apart. The target population of the trial is patients with a diagnosis of lupus undergoing rituximab therapy as standard of care. The trial began recruitment in November 2016 and closed for recruitment in March 2019, all patients were recruited in England, with data collection of the primary outcome expected to end in March 2020. Collection of safety data will end in April 2020.

### Study population

Detailed inclusion and exclusion criteria are listed in box 1. In brief, patients are aged between 18 and 75 years with four or more criteria for SLE, according to the American College of Rheumatology (ACR) 1997 criteria or Systemic Lupus International Collaborating Clinics 2012 criteria for lupus, or biopsy proven lupus nephritis with one additional supportive test on at least two occasions (positive antinuclear antibody, anti-dsDNA antibodies or anti-Smith antibodies).[14] Patients had to have a history of anti-dsDNA antibodies detectable at least once in the past 5 years. Participants had to be eligible for treatment with rituximab as per NHS England Commissioning Policy. A screening and assessment visit are performed before the first (of two) rituximab infusion, which is 4 to 8 weeks before randomisation (Day 0, figure 1). A second screen is performed no more than 10 days before randomisation to confirm eligibility (eg, excluding patients who required intravenous antibiotics for infections developing after rituximab or low IgG levels, see box 1). Prior use of rituximab is allowed, but not within 6 months of initial screening.

## Box 1    Participant inclusion and exclusion criteria (randomisation to belimumab or placebo occurs at Day 0)

### Inclusion criteria

1. Age between 18 and 75 years.
2. Participants with four or more criteria for systemic lupus erythematosus (SLE) according to the American College of Rheumatology 1997 criteria or Systemic Lupus International Collaborating Clinics 2012 criteria or biopsy proven lupus nephritis with one additional supportive test on at least two occasions (positive antinuclear antibody, anti-dsDNA antibodies or anti-Smith antibodies).
3. History of anti-dsDNA antibodies detectable at least once in the past 5 years prior to screening (ELISA test is preferable for anti-dsDNA antibody testing).
4. Participants are due to be treated with the first infusion of B cell depletion therapy (rituximab) 4 to 8 weeks before randomisation (Day 0). Previous use of rituximab is allowed prior to this cycle but not within 6 months of screening.
5. No contraindications to the use of belimumab.
6. Ability to provide informed consent.

### Exclusion criteria

1. Severe 'critical' SLE flare defined as British Isles Lupus Assessment Group A flare in central nervous system or any SLE manifestation requiring more immunosuppression than allowed within the protocol in the physician's opinion.
2. Pregnancy and/or breastfeeding patients.
3. At risk of pregnancy and unwilling to use an acceptable form of birth control contraception.
4. Prior use of belimumab, atacicept or any biological therapy (except rituximab, but no other B cell depleting therapies).
5. Participation in any other interventional trial within the last 6 months.
6. Estimated glomerular filtration rate <30 mls/min at screening.
7. Active infections, including but not limited to:
   – Current or past infection with hepatitis B or C as defined by:

Hepatitis B surface antigen positive.

Hepatitis B surface antibody positive and hepatitis B core antibody positive.

Hepatitis C antibody positive.
   – Historically positive HIV test or test positive at screening for HIV.
   – Active tuberculosis.
8. Infection history:
   – Currently on any suppressive therapy for a chronic infection (such as tuberculosis, pneumocystis, cytomegalovirus, herpes simplex virus, herpes zoster and atypical mycobacteria)
   – Hospitalisation for treatment of infection within 60 days of Day 0.
   – Use of parenteral (intravenous or intramuscular) antibiotics (antibacterials, antivirals, anti-fungals or anti-parasitic agents) within 30 days of Day 0.
   – Receipt of a live-attenuated vaccine within 3 months of Day 0.
   – In the investigator's opinion, participants that are at high risk for infection (including but not limited to in dwelling catheter, dysphagia with aspiration, decubitus ulcer, history of prior aspiration pneumonia or recurrent severe urinary tract infection).
9. IgG levels below 4.0 g/L, IgA level < 10 mg/dL (IgG and IgA test must be performed no more than 10 days before Day 0).
10. Primary immunodeficiency.
11. History of malignant neoplasm within the last 5 years.

## Box 1    Continued

12. History of cervical dysplasia cervical intraepithelial neoplasia- CIN grade III, cervical high-risk human papillomavirus (HPV) or abnormal cervical cytology other than abnormal squamous cells of undetermined significance within the past 3 years. The participant will be eligible after the condition has resolved (eg, follow-up HPV test is negative or cervical abnormality has been effectively treated >1 year ago).
13. Severe, progressive or uncontrolled renal, hepatic, haematological, gastrointestinal, pulmonary, cardiac or neurological disease or, in the investigator's opinion, any other concomitant medical condition or significant abnormal laboratory value that places the participant at risk by participating in this trial with the exception of diseases or conditions related to active SLE.
14. Comorbidities not lupus related currently requiring systemic corticosteroid therapy.
15. Evidence of serious suicide risk including any history of suicidal behaviour in the last 6 months and/or any suicidal ideation in the last 2 months or who in the investigator's judgement, pose a significant risk.
16. History of an anaphylactic reaction to parenteral administration of contrast agents, human or murine proteins or monoclonal antibodies. Current drug or alcohol abuse or dependence, or a history of drug or alcohol abuse or dependence within 364 days prior to Day 0.
17. White blood cells <1.5 x 109/L, neutrophils <1 x 109/L measured up to 10 days before Day 0.
18. A history of major organ transplant or haematopoietic stem/cell/marrow transplant or renal transplant.

### Randomisation and blinding

BEAT Lupus is a double-blind trial. Only the local pharmacist, the unblinded site monitor and the trial statistician will be unblinded to the treatment allocation. All other members of the trial team and patients are blinded to the trial medication throughout the trial. Unblinding will only occur if a participant experiences a serious adverse event (SAE) for which the clinical management of the SAE will be facilitated by the unblinding of the participant's treatment allocation. All recruited participants will be given a card with contact details for the trial team including emergency contact 24 hours a day, 7 days per week. In the event of unblinding becoming necessary, emergency unblinding can occur at any time via the 24 hours web-based service provided by Sealed Envelope (https://www.sealedenvelope.com/).

After providing written informed consent, participants are randomised to belimumab or placebo treatment using a secure online randomisation service provider, Sealed Envelope. The trial statistician will generate a list of unique BEAT Lupus infusion codes which will be sent to Sealed Envelope. The trial pharmacist will dispense the randomised infusion for each participant in accordance with their infusion code. Randomisation will be performed using minimisation, incorporating a random element to ensure balance in the stratifying variables between the two randomised groups, while retaining an element of unpredictability. Stratifying variables are: CD19

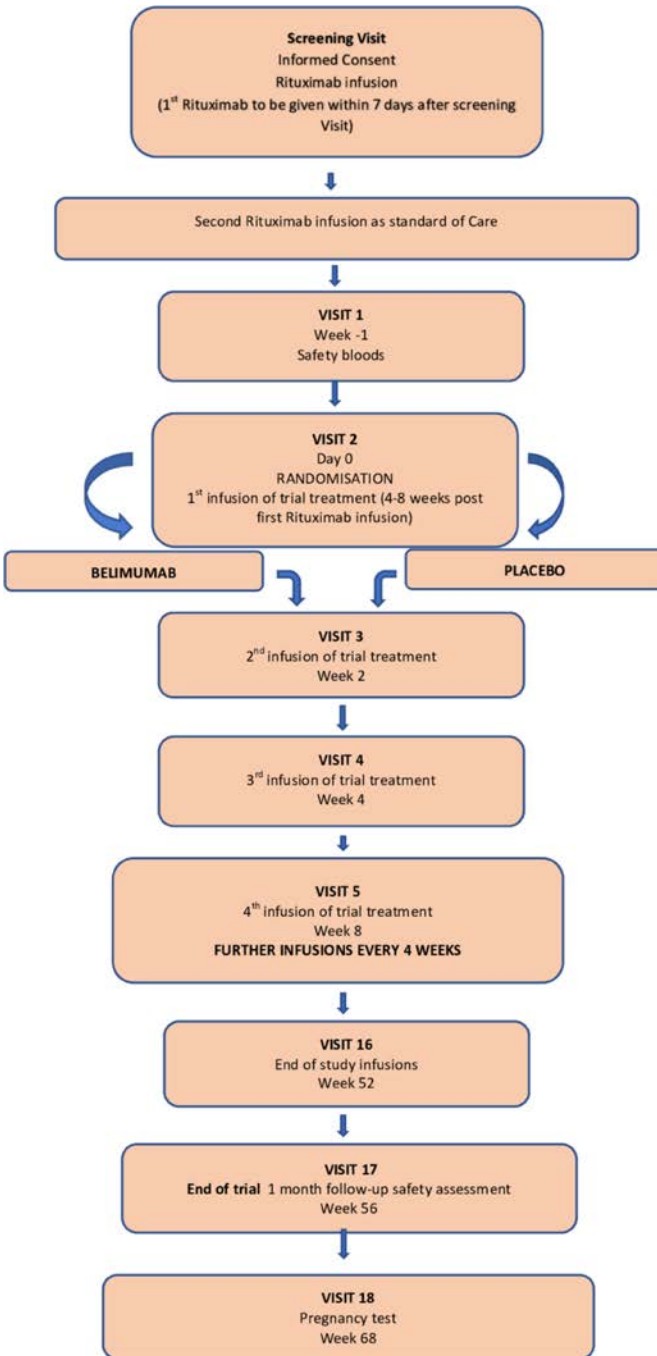

**Figure 1** Schedule of trial visits.

count after rituximab at randomisation ($<0.01\times10^9$/l vs $>0.01\times10^9$/l) to account for variability in B cell depletion which would affect response, anti-dsDNA (positive or negative at screen) and whether patients have active renal disease at screening.

## Trial treatment

Participants either receive intravenous belimumab or placebo 4 to 8 weeks after the first of two infusions of B cell depletion therapy (one cycle rituximab) administered by trial nurses. The total treatment period (belimumab or placebo) will be 52 weeks (last infusion of belimumab or placebo at week 52) followed by a safety visit at week 56.

In the active therapy arm, participants will receive belimumab according to the standard dosage regime. The recommended dosage regimen is 10 mg/kg at 2 week intervals for the first three doses and at 4 week intervals thereafter. The participants in the placebo group receive the same volume of normal saline infusions (indistinguishable from active treatment) at the same time points as the active treatment group. The visit schedule is summarised in figure 1. Trial treatment may be stopped in the event of unacceptable toxicity or adverse event.

## Concomitant medications

Before randomisation, which includes the period when rituximab is administered, treatment is entirely at the physician's discretion, though the rituximab dose is fixed by NHS England (1 g administered on two occasions, 2 weeks apart). All medication is recorded. From the day of randomisation (Day 0, see figure 1), the maximum dose of mycophenolate is 1 g/day with a suggested dose reduction to 500 mg/day at 3 months or as soon as possible thereafter. The maximum dose of azathioprine after randomisation is 1 mg/kg with a suggested dose reduction to 0.5 mg/kg by 3 months or as soon as possible thereafter. The maximum dose of methotrexate allowed is 15 mg/week with a dose reduction to 10 mg/week by 3 months or as soon as possible thereafter. Only one of these three immunosuppressant drugs (azathioprine, methotrexate and mycophenolate) can be administered post randomisation. No other immunosuppressants (eg, cyclophosphamide, tacrolimus) are allowed after randomisation except in the case of severe flare. Background anti-malarial drugs are permitted but no new anti-malarials can be started after randomisation. Mepacrine (quinacrine) is acceptable for skin problems resistant to hydroxychloroquine and both can be combined if this regime was started before randomisation. No change in dose of the anti-malarials can be made after randomisation. Participants will be permitted to receive up to 20 mg prednisolone/day from randomisation. It is strongly encouraged that prednisolone should be reduced by 50% by 6 months post randomisation, with a maximum dose of 5 mg/day recommended at 6 months if possible. Participants who flare with one British Isles Lupus Assessment Group (BILAG) A or two BILAG B scores will be permitted to receive other therapies or increased prednisolone at the discretion of the treating physician. Further cycles of rituximab are not allowed during the trial even in the event of a severe disease flare. Centres are encouraged to follow the British Society for Rheumatology guidelines for the management of systemic lupus erythematosus in adults.[14]

## Trial endpoints

The primary outcome measure is anti-dsDNA antibody levels at 52 weeks. Anti-dsDNA antibodies often correlate with disease activity.[14 26 27] This biomarker was also chosen due its correlation with BAFF levels in participants post B cell depletion therapy.[19] Serum anti-dsDNA antibodies

## Box 2 Outcome measures

### Primary outcome measures
Anti-doublestranded DNA (anti-dsDNA) antibody levels at 52 weeks.

### Secondary outcome measures
1. Anti-dsDNA antibody levels at 24 weeks.
2. Proportion of participants with any adverse events by 52 weeks.
3. Proportion of participants with any serious adverse events by 52 weeks.
4. Proportion of participants with any infections by 52 weeks.
5. Proportion of participants with a severe disease flare (defined as ≥ one British Isles Lupus Assessment Group (BILAG) A) by 52 weeks.
6. Proportion of participants with a severe (defined as ≥ one new BILAG A) or a moderate disease flare (defined as ≥ two new BILAG Bs) by 52 weeks.
7. Time to severe disease flare.
8. Proportion of participants with a severe disease flare by 24 weeks.
9. Systemic Lupus Erythematosus Disease Activity Index 2000 at 52 weeks.
10. Subject Global Assessment of Disease Activity at 52 weeks.
11. C3 levels at 52 weeks.
12. Immunoglobulin levels at 52 weeks.
13. Cumulative steroid dose from randomisation to 52 weeks.
14. Proportion of participants decreasing steroid dose at randomisation by 50% without experiencing a flare, or if below 10 mg/day at randomisation reducing dose to 5 mg/day or less at 52 weeks.
15. Lupus Quality of Life, SF-36 at 52 weeks
16. Columbia Suicide Severity Rating Scale to assess suicidality risk at 52 weeks.

### Exploratory outcomes
1. B cell activating factor (measurement of RNA from whole blood) at 52 weeks.
2. Kinetics of B cell repopulation.

are analysed using the DIASTAT anti-dsDNA ELISA. All samples are tested in the same central lab at University College London (UCL).

Secondary outcome measurements include anti-dsDNA antibody levels at 24 weeks and the proportion of participants with serious adverse events (box 2). The proportion of participants with a moderate (defined as ≥2 new BILAG Bs) or severe disease flare (defined as ≥1 new BILAG A) by 24 and 52 weeks following randomisation will be assessed, as will the time to a disease flare. Patients who have a severe disease flare will be invited to an additional visit where standard trial data, including serum for the primary endpoint analysis, is collected, particularly if they are withdrawing from trial therapy and may be treated with therapies that are not allowed according to the protocol for example, rituximab and cyclophosphamide.

Disease activity will be assessed using the British Isles Lupus Assessment Group (BILAG-2004) disease activity index[28] and Systemic Lupus Erythematosus Disease Activity Index 2000.[29] C3 and immunoglobulin levels will be recorded. As part of the assessment of response to treatment, we shall calculate the cumulative steroid dose during treatment from randomisation and the proportion of participants decreasing their steroid dose by 50%

without flaring from randomisation; or if below 10 mg/day at randomisation reducing steroid dose to 5 mg/day or who discontinue glucocorticoids with stable disease. Lupus Quality of Life and SF-36 will be used as patient reported outcome measures. ColumbiaSuicide Severity Rating Scale will be used to assess suicidality risk.

### Data collection and management
All participant data will be collected by members of the clinical trial team as described in the Delegation Log. Data will be recorded on a paper Case Report Forms and entered in to the secure Clinical Trial Database via a secure web interface. All data will be handled in accordance with the Data Protection Act 1998. The participant's initials, date of birth and participant identification number, will be used for identification.

Data entry, coding and security will be the responsibility of a dedicated data manager based. A custom designed database will be created (using InferMed's MACRO V4) to store all trial data. This will be compliant with all necessary regulatory requirements including an audit trail to record all date/time stamped corrections accompanied by justification/explanation for any data amendments. Monitoring visits will be conducted by the central trial team at all recruiting sites at regular intervals, in accordance with regulatory requirements. The trial may be audited independently by the Medicines & Healthcare Products Regulatory Agency.

### Trial sample size calculation
We aim to recruit up to 56 SLE participants (the original sample size of 50 was increased to account for a higher than expected dropout rate) and randomise 28 participants to receive belimumab after rituximab and 28 participants to receive placebo after rituximab. The primary outcome is the estimated difference in anti-dsDNA binding antibody levels at 12 months, adjusted for anti-dsDNA binding levels at screening. Sample size calculations were performed using Stata Release 13 assuming that anti-dsDNA binding antibody levels are log normally distributed. The SD of anti-dsDNA and the correlation structure assumed were based on two sets of data (a) the study of 35 participants by Carter et al[19] and (b) unpublished data provided by Prof D Isenberg for 67 participants before and 6 months after B cell depletion therapy (table 1).

On the natural log scale the SD of anti-dsDNA at the end of treatment was taken to be 1.7 based on the above data. The correlation between before rituximab and final log anti-dsDNA was taken to be 0.55 also based on the above. Twenty-two evaluable participants per group would be able to detect a difference of 1.2 in anti-dsDNA binding antibody levels on the natural log scale at the 5% significance level with 80% power. Assuming that 20% of participants fail to attend the 52 week follow-up visit, and taking account of the correlation between screening measurements and 52 week measurements, we aim to recruit 28 participants per group. For example, if we assume that

**Table 1** SD and correlations for log$_e$ anti dsDNA

| Data set | Time point | Log mean | Log SD | Correlation |
|---|---|---|---|---|
| Isenberg (n=67) | Before rituximab | 6.036 | 1.393 | |
| | Follow-up | 5.189 | 1.500 | 0.612 |
| Carter (n=35) | Before rituximab | 5.091 | 1.676 | |
| | Follow-up | 5.225 | 1.781 | 0.527 |

there will be no change in anti-dsDNA levels in the belimumab group, then we would be able to detect an increase of 232% in the placebo group as significantly different. To place this increase in context, Carter *et al* showed that the difference in anti-dsDNA binding antibody levels on the natural log scale between those participants who did and did not have a flare was 1.928, corresponding to a 588% increase in anti-dsDNA binding levels in participants with a disease flare.[19]

### Statistical analysis

A Consolidated Standards of Reporting Trials diagram will be used to describe the course of participants through the trial (figure 1). Characteristics at screening and randomisation will be summarised by treatment group.

The analysis of the primary outcome will be done using linear regression to estimate the difference in log anti-dsDNA levels between the two groups at 52 weeks, adjusting for the stratifying variables anti-dsDNA (at screening, pre-rituximab), BILAG category (renal, non-renal) (at screening, pre-rituximab) and CD19 count (at randomisation). The analysis will be done on the intention-to-treat principle; all available 52 week measurements will be included regardless of patient's adherence to their randomised treatment. If withdrawals from follow-up before week 52 exceed 10%, a supportive analysis of the primary outcome will be done using multiple imputation (with 52 week anti-dsDNA imputed using baseline and any 12 and 24 weeks measurements where available and other auxiliary information) to evaluate the sensitivity of the results to any differential loss to follow-up. Another supportive sensitivity analysis will be done to estimate the treatment effect for only those patients who adhered to their allocated treatment for 52 weeks. Additionally, if material differences are found between treatment groups in cumulative prednisolone dose from randomisation (p<0.1), mediation analyses will be done to evaluate whether the observed effect of belimumab on anti-dsDNA levels is mediated by changes in prednisolone dose.

A similar modelling approach will be adopted for the analysis of other continuous outcomes measured at screening (before rituximab) and at 52 weeks. Fisher's exact test will be used to compare proportions. Two sample t-tests or Mann-Whitney U tests, depending on the distribution of the data, will be used to evaluate continuous outcomes compared only at the end of the trial. Kaplan-Meier survival curves will be produced for comparisons of time to an event. The statistical analysis will be performed using Stata.

### Ethical considerations

The trial will be conducted in accordance with ethical principles derived from Good Clinical Practice Guidelines, including the Declaration of Helsinki. Written informed consent (see online supplementary file for consent form) will be collected from all patients by an approved study nurse or physician before the start of any study procedures. Patients will be assigned a participant identification number and all patient information transferred to the sponsor will contain this number, patient initials and date of birth. All SAEs will be reported to the sponsor within 24 hours. All agreed substantial protocol amendments, will be documented and submitted for ethical and regulatory approval prior to implementation and communication to sites. Protocol V.5 dated 21st January 2019 is the current protocol version and has been approved.

### Dissemination

Trial information is available at https://www.isrctn.com/ISRCTN47873003 including study sites all of which are in England, and the results of this trial will be submitted for publication in relevant peer-reviewed publications. Key findings will also be presented at national and international conferences. Published results will be disseminated to investigators at participating sites, who will further disseminate the results to trial participants on request. Authorship of publications will adhere to International Committee of Medical Journal Editors guidelines.

### Patient and public involvement

Feedback gathered from focused patient involvement meetings was used to inform the design of this trial. Patients recognised the need for improving therapies for SLE. Our patient partners suggested that the inclusion and exclusion criteria should not represent a significant barrier to recruitment and that the trial should be as close to 'real-world' practise as possible. Two patient advisors are members of the Trial Steering Committee.

### Trial steering committee

The Trial Steering Committee (TSC) is led by independent chair (Professor Raashid Luqmani) from a non-participating centre, and includes an independent statistician (Professor John Bankart) and one independent clinical expert in SLE (Dr Ceril Rhys-Dillon) as well as the Chief Investigator (Professor Michael Ehrenstein) and members of the trial team, some principal investigators from the participating sites and two patient partners. The TSC provides expert oversight of the trial, agreement of final protocol, reviewing progress of study, recruitment and retention rates and agreeing changes to the protocol to facilitate the smooth running of the study. The TSC together with the Chief Investigator will oversee the

timely analysis and publication of the main trial results and will review requests for data sharing. Full composition of the TSC and terms of reference are available from the corresponding author.

## Data monitoring committee

An Independent Data Monitoring Committee (IDMC) is chaired by Professor Richard Watts (independent clinical expert) and includes Professor Vern Farewell (independent statistician) and Professor Robert Moots (independent clinical expert) together with the trial statistician. During the period of recruitment to the study, interim analyses of accruing data will be supplied, in strict confidence, to the IDMC. The IDMC will advise the TSC of any issues to safeguard the interests of trial participants including assessment of SAEs and flare rates after 50% of the participants have been recruited and followed up for 6 months to determine whether recruitment should continue (interim safety analysis). All SAEs will be reported in real time to the IDMC with a clinical narrative, including an assessment of attribution to belimumab. Full composition of the IDMC and terms of reference are available from UCL Comprehensive Clinical Trials Unit.

## DISCUSSION

This clinical trial was developed through investigation and analysis of clinical and laboratory data of lupus patients receiving rituximab. In particular, we noted increased serum BAFF levels correlated with a rise in anti-DNA antibodies at the time of flare in lupus patients post rituximab.[19] We therefore hypothesised that lupus patients treated with belimumab following rituximab therapy will have lower anti-dsDNA antibody levels compared with those that have received rituximab alone, and that this combination would have greater efficacy than rituximab alone. Given the paucity of novel therapies, and the many clinical trials that have failed to show efficacy for SLE, a combination of biological therapies with complementary effects given in succession may be required to control disease. If the results of this trial are promising, a larger trial will be required of sufficient power to detect improved clinical outcomes. Indeed, a larger trial is already underway (BLISS BELIEVE NCT03312907) testing whether rituximab given after belimumab confers an additional benefit compared with belimumab alone.[30] The order of administration of these two biologics in BLISS BELIEVE is reversed compared to BEAT LUPUS, and there is no rituximab alone arm. The administration of rituximab first allows for the detection of adverse events that occur in the 4 to 8 week period after rituximab and before randomisation. A second screen 7 to 10 days before randomisation will exclude patients based on these adverse events. CALIBRATE (NCT02260934), a clinical trial in lupus nephritis uses a similar sequence of these two biologics to the trial reported here, though cyclophosphamide is administered with the rituximab,

and higher doses of concomitant immunosuppressive medications are allowed.

Recruitment to lupus trials has been challenging in the UK, and few academic led interventional SLE trials in the UK have met their recruitment target. The last academic led lupus trial in the UK not subject to early termination because of poor recruitment commenced in 1997 and completed recruitment in 2005 (though even this open label trial's recruitment target had to be reduced due to slow enrolment).[31] We endeavoured to optimise recruitment by designing a protocol that would be as close to real life clinical practice as possible. Consequently, there are no therapeutic restrictions before patients are randomised (to receive either belimumab or placebo) giving freedom to the treating physician including the possibility of administering high dose corticosteroids in conjunction with rituximab. Rituximab had to be administered 4 to 8 weeks before randomisation and is administered in accordance with the NHS England's Interim Commissioning Policy (NHS ENGLAND A13/PS/a) for patients with refractory lupus.[14] This policy only allows rituximab to be given to patients who have failed to respond adequately to either mycophenolate or cyclophosphamide, unless contraindicated, or requiring unacceptably high doses of corticosteroids. Thus, the results of this trial will be relevant to lupus patients that are refractory or intolerant to conventional treatment, the patient group for which this combination therapy is most likely to be funded. The study provides information about rates of recruitment to lupus trials in England which could inform the development and feasibility of other UK based trials planned in the future. Recruitment was aided through NHS England's policy applicable to biological therapies for SLE which encourages physicians and their patients to consider interventional clinical trials (NHS ENGLAND A13/PS/a).[14]

The trial protocol offers physicians some flexibility with the use of concomitant immunosuppressant and oral prednisolone after randomisation. However, because of safety concerns, and to maximise the chances of revealing any potential difference between the two treatment arms, corticosteroids are limited to a maximum dose of 20 mg/day at randomisation and it is strongly encouraged to taper the dose by at least 50% by 6 months. Moreover, immunosuppressant doses are also restricted. It can be difficult to achieve reductions in both corticosteroids and immunosuppressants in patients who, by the nature of the inclusion criteria, are more resistant to therapy. Our strategy is to advise that reduction in corticosteroid dose should be prioritised over tapering immunosuppressants due to the former's more problematic side effects, and its ability to mask the difference between treatment arms in lupus trials.

This trial has some limitations. First, the sample size is relatively small because at the time of the trial's inception there were no published safety data on the combination of rituximab and belimumab and an earlier phase, smaller trial was advised. Therefore, the trial is powered

on anti-DNA antibodies rather than a clinical endpoint. To limit the risk of adverse events due to immunosuppression, the dose of concomitant immunosuppressive medication is lower than routinely prescribed for patients with active disease, particularly renal patients receiving mycophenolate. This may reduce recruitment of patients with active nephritis by some physicians, although this restriction in dosage could also be an advantage with respect to distinguishing between active drug and placebo. The serological inclusion criteria enable patients with a confirmed positive dsDNA antibody within the last 5 years to participate in the trial rather than patients who have raised anti-DNA antibodies at screen. This approach was taken to maximise recruitment but also with the recognition that flares post rituximab can, in a proportion of patients, be associated with increased anti-DNA antibody levels above that observed when rituximab is first commenced.[15]

The development of this trial has arisen directly from analysis of the immunological changes following B cell depletion therapy. Building on this experimental medicine approach, analysis of B cell phenotype has been incorporated into this trial to understand how sequential targeting of B cells and BAFF affect B cell numbers and phenotype. If the overall results from this trial are encouraging, a larger definitive trial using time to flare as the primary outcome may be considered after taking into account the results from other similar trials currently underway. The increasing interest in this combination therapy has the potential to offer a novel treatment option for patients with active lupus.

**Author affiliations**
¹Centre for Rheumatology, Division of Medicine, University College London, London, UK
²Comprehensive Clinical Trials Unit, University College London, London, UK
³Rheumatology Research Team - Inflammation and Ageing (IIA), University of Birmingham Research Laboratories, New Queen Elizabeth Hospital, Birmingham, UK

**Acknowledgements** We acknowledge the important contribution of the BEAT Lupus Trial Steering Committee, the Data Monitoring Committee and all the patients participating in the trial or involved in its development. All authors reviewed and edited the draft version of the manuscript and approved the final version submitted. We are indebted to the BEAT Lupus Trial Collaborators at the recruiting sites, the British Isles Lupus Assessment Group (BILAG) and The NIHR Musculoskeletal Translational Research Collaboration for their advice and support. The trial is sponsored by University College London, Gower Street, London, WC1E 6BT, UK, +44 20 7679 6163, ctu.beatlupus@ucl.ac.uk.

**Contributors** MRE conceived of the study. MRE, CG, CJD and DAI initiated the study design. PM and KC and CJD provided statistical expertise in clinical trial in clinical trial design. MRE, CG, CJD, FI and EC assisted with protocol development. MRE, DAI, CG and CJD are grant holders.

**Funding** This trial is supported by Versus Arthritis (grant number 20873) and the UCLH Biomedical Research Centre (BRC). GSK are providing belimumab free of charge, as well as additional funding. The MRC (MASTERPLANS CONSORTIUM) is supporting some of the experimental medicine applied to samples from this trial. GSK had no role in the design of this study and will not have any role during its execution, analyses, interpretation of the data or decision to submit results. Versus Arthritis and the MRC reviewed the relevant grant proposals and monitor progress of relevant aspects of the study but will not play any role in the analyses, interpretation of data or decision to submit results.

**Competing interests** MRE has received grant/research support from GSK. CG has been a member of the speakers bureau for GSK and has received consultancy fees for attending advisory boards.

**Patient consent for publication** Not required.

**Ethics approval** London - Hampstead Research Ethics Committee REC reference: 16/LO/1024, Health Research Authority IRAS project ID: 195085, EudraCT number: 2015-005543-14.

**Provenance and peer review** Not commissioned; externally peer reviewed.

**ORCID iDs**
Patrick Muller http://orcid.org/0000-0002-6824-578X
David A Isenberg http://orcid.org/0000-0001-9514-2455
Michael R Ehrenstein http://orcid.org/0000-0003-1673-743X

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
