## [Reviewer comments · BMJ Open]

ARTICLE DETAILS

TITLE (PROVISIONAL)	Belimumab after B cell depletion therapy in patients with systemic lupus erythematosus (BEAT LUPUS) protocol: A prospective multi-centre, double-blind, randomised, placebo-controlled, 52-week phase II clinical trial
AUTHORS	Jones, Alexis; Muller, Patrick; Dore, Caroline; Ikeji, Felicia; Caverly, Emilia; Chowdhury, Kashfia; Isenberg, David; Gordon, Caroline; Ehrenstein, MR

VERSION 1 – REVIEW

REVIEWER	Peter Watson University of Cambridge UK
REVIEW RETURNED	09-Jul-2019

GENERAL COMMENTS	Belimumab after B cell depletion therapy in patients with systemic lupus erythematosus (BEAT LUPUS) trial protocol: A prospective multicentred, double-blind, randomised, placebo-controlled, 52-week phase II clinical trial bmjopen-2019-032569 Page 7, line 6. It states this is a multicentre trial. Do you intend investigating differences in patients across centres to ensure homogeneous outcomes across centres? Page 7, line 54. Please state and reference which minimisation software you are using. Page 8, lines 30-32. What does "treatment is entirely at the physician's discretion" mean in practice? Is this saying it is up to the physicians how strong a dose of Rituximab is administered and how often in the baseline period prior to randomisation with patients only given Rituximab if their CD19 counts become too high (?) rather than at fixed intervals on fixed doses? If so, and following on from this, I wonder how standard it is for the baseline period to involve people who are already receiving a pre-existing treatment (a drug such as Rituximab) and, as a result, are being treated with possibly different frequencies and strengths of dose of this drug or indeed with "other therapies" (page 9, line 25). Could those receiving more intensive treatment with Rituximab or other therapies, for example, perhaps have more underlying problems which might additionally be needed to be taken into account in the analysis comparing the treatment and placebo groups? With so many apparently disparate approaches being used on the patients at baseline to keep them "healthy" I wonder if we really know the natural state of the patients (ie heir true state at
---

	baseline) which one might argue is required information prior to the randomisation. Page 10, line 5-15. It is not clear to me to what the (logged) difference of 1.2 used in the power calculation refers. I believe the effect size of 232% mentioned on line 7 must represent the difference BETWEEN groups, which are what you are seeking to compare, rather than within groups as stated. Please, therefore, clarify and define which effect is being used in the power calculation. The most sensible effect would be the group x time interaction comparing the difference between baseline and week 52 logged anti-dsDNA between the groups. If so, I find you are sufficiently powered even for smaller correlations than the one (0.55 on line 12) that you mention. I think, however, the difference you mention on page 10, line 7 is large. Indeed as you claim this amounts to an increase of 232% (a trebling of the baseline value at 52 weeks) which seems very large. Please, therefore, justify where this log difference of 1.2 comes from and explain why detecting smaller differences is not of interest. I also think the stated unreferenced expected correlation of 0.55 (line 12) between baseline and a measure taken a year later seems overly high. I wonder similarly, therefore, how you would justify expecting to find such a high correlation. Expecting too high a difference and too high a correlation could lead to underpowering of the study. Page 10, lines 30-50. I am not clear how you would account for the expected 20% (stated on page 10, line 14) of dropouts in the intention-to-treat analysis that you propose on "available" 52 week measurements (line 32). None of the statistical analyses stated per se include missing values. You could, for example, use multiple imputation or mixed models to include information from dropouts. You should also state which software you are intending to use in these analyses.
--	--

REVIEWER	Julia Simard Stanford Medicine
REVIEW RETURNED	30-Jul-2019

GENERAL COMMENTS	The authors present and summarize a protocol for BEAT LUPUS - an RCT for Belimumab treatment following Rituximab. As they mention in the manuscript, there are other studies that have evaluated these therapies, some also in combination or following one another (but in a different order). The rationale for this approach appears solid. There were a number of questions that came up during the review of this manuscript that warrant clarification and will greatly help readers with respect to understanding the study and also facilitate reproduction in future work.  1. Please clarify early on, including in the abstract, that there are two Rituximab infusions following the screening visit and before randomization (according to Figure 1). There are several parts of the manuscript where the reader may misinterpret that it is just one dose. 2. Abstract, Methods and analysis section, line 25, please consider changing "receiving Rituximab" to "initiating" or "starting Rituximab"
---

	so that it is clear from the start that these are incident Rituximab users. 3. The authors refer to "standard of care" throughout the manuscript but it would be of benefit to understand what this is at their institution. Some may find the referring to such a standard as controversial as there are no clear universal guidelines on this to date. 4. To better understand the eligibility and the interpretation, please clarify what previous rituximab use was ok in this study. What there a washout? What duration? This criterion is in Box #1 but needs more detail to understand the potential study population and the generalizability. 5. Please make sure that the use of baseline, at randomization, screening, etc. are clear and that they are necessary. 6. Need more clarity on the research question/hypothesis. The outcome appears to be change in dsDNA - among those without dsDNA antibodies when the labs were checked at the start of the study (but a history of dsDNA antibodies) how were they factored in? Also, based on the description in the sample size calculation section it appears that there is an anticipated increase in antidsDNA levels but intuitively sometimes it seems that there should be a decrease if the treatment is working (e.g. anstract, last sentence of the methods/analysis suggests a target of lower antidsDNA antibodies? Can the authors clarify in the protocol the the precise research question/hypothesis is around the change, direction, and how it will be estimated (pre-post vs post-pre, for example)? 7. Why was there a pregnancy test at week 68? This is mentioned in the text and also shown in the figure as part of the protocol but seems perhaps like an unnecessary detail. 8. The authors note that the power is limited and based on the antidsDNA endpoint, however describe a number of analyses including concomitant medications following randomization as potential mediators. In theory this is an interesting question but it seems very unlikely that there will be sufficient power to assess this.
--	--

REVIEWER	Peter H Schur MD Brigham and Women's Hospital Harvard Medical School USA
REVIEW RETURNED	04-Aug-2019

GENERAL COMMENTS	1. If anti-dsDNA such an essential outcome, having an anti-dsDNA in the last 5 years as an entry criteria, is inadequate--would make a pos antidsDNA assay an entry criteria within the last 2 weeks 2. Anti-dsDNA assays vary in sensitivity and specificity and association with active disease. Need to state what assay used, and make sure that each patient is tested with the same assay, in the same lab 3. Introducing other immunosuppressives such as mycophenolate, azathioprine, methotrexate, or variable doses of corticosteroids, introduces a variable that will be hard to sort out given the small size of the study 4. This is a very small study, considering all the variables 5. anti-dsDNA is a ok endpoint, but looking at renal disease (GFR,
--

urinary sediment, urine protein is critical to assess how good/effective a therapeutic plan is
--

VERSION 1 – AUTHOR RESPONSE

Reviewer: 1

Reviewer Name: Peter Watson

Institution and Country: University of Cambridge UK

Please state any competing interests or state 'None declared': None declared

Please leave your comments for the authors below Belimumab after B cell depletion therapy in patients with systemic lupus erythematosus (BEAT LUPUS) trial protocol: A prospective multicentred, double-blind, randomised,

placebo-controlled, 52-week phase II clinical trial bmjopen-2019-032569

Page 7, line 6. It states this is a multicentre trial. Do you intend investigating differences in patients across centres to ensure homogeneous outcomes across centres?

We intend to recruit 56 patients from 16 centres, so on average there will be 3 patients per centre. Unfortunately, there will not be enough patients at each centre to assess homogeneity of outcomes across centres.

Page 7, line 54. Please state and reference which minimisation software you are using.

After providing written informed consent, participants are randomised to Belimumab or placebo treatment using a secure online randomisation service provider, Sealed Envelope <https://www.sealedenvelope.com/> . This has been amended on the manuscript on page 6.

Page 8, lines 30-32. What does "treatment is entirely at the physician's discretion" mean in practice? Is this saying it is up to the physicians how strong a dose of Rituximab is administered and how often in the baseline period prior to randomisation with patients only given Rituximab if their CD19 counts become too high (?) rather than at fixed intervals on fixed doses?

If so, and following on from this, I wonder how standard it is for the baseline period to involve people who are already receiving a pre-existing treatment (a drug such as Rituximab) and, as a result, are being treated with possibly different frequencies and strengths of dose of this drug or indeed with "other therapies" (page 9, line 25). Could those receiving more intensive treatment with Rituximab or other therapies, for example, perhaps have more underlying problems which might additionally be needed to be taken into account in the analysis comparing the treatment and placebo groups? With so many apparently disparate approaches being used on the patients at baseline to keep them "healthy" I wonder if we really know the natural state of the patients (ie heir true state at baseline) which one might argue is required information prior to the randomisation.

The only rituximab treatment regime funded (and recommended) by NHS England for patients with SLE is to give two 1g doses of rituximab two weeks apart. Thus, all patients will receive the same dose of rituximab. We have referred to the NHS England Interim Commissioning Guidelines on page 6 of the manuscript as well as referencing the British Society for Rheumatology guideline for the management of systemic lupus erythematosus in adults (reference 14).

Rituximab dose does not change according to CD19 measurements after the first dose of rituximab (i.e. before randomisation). After randomisation, no further infusions of rituximab are allowed. We are using CD19 as a stratifying variable at randomisation to ensure that there is an equal balance of patients who have fully B cell depleted and those that have undergone only partial B cell depletion in the placebo and active treatment arm.

Four to eight weeks before randomisation patients receive their first (of two) infusion of rituximab. All other treatments are recorded and are according to standard practice (reference 14) thereby reflecting real life clinical practice which will have greater clinical applicability. Data about patient treatment and disease activity are collected before rituximab and will be available for comparison when the results are analysed.

Page 10, line 5-15. It is not clear to me to what the (logged) difference of 1.2 used in the power calculation refers. I believe the effect size of 232% mentioned on line 7 must represent the difference BETWEEN groups, which are what you are seeking to compare, rather than within groups as stated. Please, therefore, clarify and define which effect is being used in the power calculation. The most sensible effect would be the group x time interaction comparing the difference between baseline and week 52 logged anti-dsDNA between the groups. If so, I find you are sufficiently powered even for smaller correlations than the one (0.55 on line 12) that you mention. I think, however, the difference you mention on page 10, line 7 is large. Indeed as you claim this amounts to an increase of 232% (a trebling of the baseline value at 52 weeks) which seems very large. Please, therefore, justify where this log difference of 1.2 comes from and explain why detecting smaller differences is not of interest.

I also think the stated unreferenced expected correlation of 0.55 (line 12) between baseline and a measure taken a year later seems overly high. I wonder similarly, therefore, how you would justify expecting to find such a high correlation.

Expecting too high a difference and too high a correlation could lead to underpowering of the study.

The primary outcome is the estimated difference in anti dsDNA binding antibody levels at 12 months, adjusted for anti dsDNA binding levels at screening . Sample size calculations were performed using Stata Release 13 assuming that anti dsDNA binding antibody levels are log normally distributed.

The Standard Deviation of anti dsDNA and the correlation structure assumed were based on two sets of data (a) the study of 35 participants by Carter et al (Carter et al 2013) and (b) data provided by David Isenberg for 67 participants before and 6 months after B cell depletion therapy.

Table: Standard deviations and correlations for loge anti dsDNA

Dataset	Time point	Log mean	Log sd	Correlation
Isenberg (n=67)	Baseline (prior to rituximab)	6.036	1.393	
	Follow up	5.189	1.500	0.612
Carter (n=35)	Baseline	5.091	1.676	
	Follow up	5.225	1.781	0.527

On the natural log scale the standard deviation of anti dsDNA at the end of treatment was taken to be 1.7 based on the above data. The correlation between baseline and final log anti dsDNA was taken to be 0.55 also based on the above. 22 evaluable participants per group would be able to detect a difference of 1.2 in anti dsDNA binding antibody levels on the natural log scale at the 5% significance level with 80% power. Assuming that 20% of participants fail to attend the 52 week follow-up visit, and taking account of the correlation between screening measurements and 52 week measurements, we aim to recruit 28 participants per group.

For example, if we assume that there will be no change in anti dsDNA levels in the Belimumab group, then we would be able to detect an increase of 232% in the placebo group as significantly different.

To place this increase in context, Carter et al 2013 showed that the difference in anti dsDNA binding antibody levels on the natural log scale between those participants who did and did not have a flare was 1.928, corresponding to a 588% increase in anti dsDNA binding levels in participants with a disease flare.

We have added these details on page 9 of the manuscript.

Page 10, lines 30-50. I am not clear how you would account for the expected 20% (stated on page 10, line 14) of dropouts in the intention-to-treat analysis that you propose on "available" 52 week measurements (line 32). None of the statistical analyses stated per se include missing values. You could, for example, use multiple imputation or mixed models to include information from dropouts.

You should also state which software you are intending to use in these analyses.

The main analysis will be conducted according to the intention-to-treat principle, and therefore patients who withdraw from their trial treatment but are willing to provide 52-week measurements will be included in the primary analysis. Missing covariate data are not anticipated since covariates must be recorded to allocate treatment. Sensitivity analysis will be done to evaluate the impact of differential loss-to-follow-up between treatment arms amongst patients who withdraw from the trial completely prior to 52 weeks follow up. We have added a sentence describing this on page 10 of the manuscript. The statistical analyses will be performed using Stata. These details are provided on page 10 of the manuscript.

Reviewer: 2

Reviewer Name: Julia Simard

Institution and Country: Stanford Medicine

Please state any competing interests or state 'None declared': None declared

Please leave your comments for the authors below

The authors present and summarize a protocol for BEAT LUPUS - an RCT for Belimumab treatment following Rituximab. As they mention in the manuscript, there are other studies that have evaluated these therapies, some also in combination or following one another (but in a different order). The rationale for this approach appears solid.

There were a number of questions that came up during the review of this manuscript that warrant clarification and will greatly help readers with respect to understanding the study and also facilitate reproduction in future work.

1. Please clarify early on, including in the abstract, that there are two Rituximab infusions following the screening visit and before randomization (according to Figure 1). There are several parts of the manuscript where the reader may misinterpret that it is just one dose.

We have clarified the dose/regime of rituximab in the abstract (page 2) and within the manuscript.

2. Abstract, Methods and analysis section, line 25, please consider changing "receiving Rituximab" to "initiating" or "starting Rituximab" so that it is clear from the start that these are incident Rituximab users.

We have changed the wording to commencing one cycle of rituximab therapy (abstract). We have clarified that patients can have previous cycles of rituximab, but not within 6 months of first screening, page 6 of the manuscript.

3. The authors refer to "standard of care" throughout the manuscript but it would be of benefit to understand what this is at their institution. Some may find the referring to such a standard as controversial as there are no clear universal guidelines on this to date.

Restrictions regarding therapy are described in the Concomitant Medication (page 7) section of the manuscript principally related to limiting and subsequently reducing dosage of immunosuppressants and steroids to prevent masking of the difference between placebo and belimumab. Only one of three disease modifying drugs (Azathioprine, methotrexate and mycophenolate) can be administered post randomisation. Apart from these restrictions centres are encouraged to follow the British Society for Rheumatology guideline for the management of systemic lupus erythematosus in adults (reference 14).

4. To better understand the eligibility and the interpretation, please clarify what previous rituximab use was ok in this study. What there a washout? What duration? This criterion is in Box #1 but needs more detail to understand the potential study population and the generalizability.

Prior rituximab use was allowed but not within 6 months of initial screening, we have stated this on page 6 and added this to Box 1 (inclusion criteria 4).

5. Please make sure that the use of baseline, at randomization, screening, etc. are clear and that they are necessary.

We have now removed the term baseline when referring to the trial time points and used screening (before Rituximab) and randomisation to ensure consistency throughout the manuscript.

6. Need more clarity on the research question/hypothesis. The outcome appears to be change in dsDNA - among those without dsDNA antibodies when the labs were checked at the start of the study (but a history of dsDNA antibodies) how were they factored in? Also, based on the description in the sample size calculation section it appears that there is an anticipated increase in antidsDNA levels but intuitively sometimes it seems that there should be a decrease if the treatment is working (e.g. abstract, last sentence of the methods/analysis suggests a target of lower antidsDNA antibodies? Can the authors clarify in the protocol the the precise research question/hypothesis is around the change, direction, and how it will be estimated (pre-post vs post-pre, for example)?

The safety of the combination (rituximab and belimumab) in patients with SLE was the foremost question imposed upon us by the trial funder Arthritis Research UK (now called VersusArthritis). At the time this study was approved, there were no relevant publications regarding safety (to date there is only a small single arm study).

We have previously shown that patients who flare after rituximab therapy can increase their anti-dsDNA antibody levels above the value observed before that cycle of rituximab (reference 19 of the manuscript). We performed a two sided sample size calculation, so did not pre-specify the direction of change. However, we expect to see higher levels of dsDNA in patients who relapse, so we would expect to see higher levels in the placebo group at 52 weeks. Hence the statement now included on page 9 in the expanded sample size calculation section "For example, if we assume that there will be no change in anti dsDNA

levels in the Belimumab group, then we would be able to detect an increase of 232% in the placebo group as significantly different. "

We therefore allowed patients who had normal levels of anti -dsDNA antibodies at screening if they had previously had elevated anti-dsDNA antibodies in the last 5 years. This decision to allow patients who did not have elevated anti-dsDNA antibodies at screening to be recruited was also made in light of the poor recruitment for this trial in its early stages (and also the poor recruitment record in the UK for clinical trials in SLE) to ensure we achieved the recruitment target.

7. Why was there a pregnancy test at week 68? This is mentioned in the text and also shown in the figure as part of the protocol but seems perhaps like an unnecessary detail.

We have removed this from the manuscript

8. The authors note that the power is limited and based on the antidsDNA endpoint, however describe a number of analyses including concomitant medications following randomization as potential mediators. In theory this is an interesting question but it seems very unlikely that there will be sufficient power to assess this.

A causal mediation analysis will also be done to quantify the extent to which any differences in anti-dsDNA at 52 weeks identified in the primary analysis are mediated by changes in prednisolone dose following randomisation. This will be viewed as a supportive, exploratory analysis. We have added this to the manuscript, page 10.

Reviewer: 3

Reviewer Name: Peter H Schur MD

Institution and Country: Brigham and Women's Hospital

Harvard Medical School

USA

Please state any competing interests or state 'None declared': None declared

Please leave your comments for the authors below

1. If anti-dsDNA such an essential outcome, having an anti-dsDNA in the last 5 years as an entry criteria, is inadequate--would make a pos antidsDNA assay an entry criteria within the last 2 weeks.

Please see response to reviewer 2 point 6. In summary, we have published data demonstrating that anti-dsDNA antibodies can increase above that seen pre Rituximab (reference 19 of the manuscript) and we are eager to explore whether this finding is replicated in the trial described here. Of relevance, this trial is principally a safety study rather than an efficacy study.

2. Anti-dsDNA assays vary in sensitivity and specificity and association with active disease. Need to state what assay used, and make sure that each patient is tested with the same assay, in the same lab

Anti-dsDNA antibodies are analysed using the DIASTAT® anti-dsDNA enzyme-linked immunosorbent assay. All samples are tested in the same central lab at UCL. This has been added to the manuscript on page 8.

3. Introducing other immunosuppressives such as mycophenolate, azathioprine, methotrexate, or variable doses of corticosteroids, introduces a variable that will be hard to sort out given the small size of the study

We will use a number of statistical approaches, for instance looking at cumulative dosages of prednisolone throughout the study to compare the variation between patients, and doing a mediation analysis to evaluate whether differences in anti-dsDNA at 52 weeks between arms were mediated by changes in concomitant medications following randomisation (mentioned on page 10 of the manuscript). We were keen to reduce any barrier to recruitment as much as possible by allowing those immunosuppressants to be used, as well as reflect real world practice. No other immunosuppressant was allowed e.g. cyclophosphamide during the study. Given that safety was a key concern we did not want to restrict the use of the commonly used immunosuppressants, and this also helped to ensure the results reflected standard clinical practice.

4. This is a very small study, considering all the variables

This was principally designed as a safety study, but given the investment by funders and patient participants we wanted to maximise the information obtained. We acknowledge that this is a small study in the limitations box and in the Discussion. This is the first RCT in lupus in the UK to complete recruitment for over a decade and limiting its sample size was a key component to ensure we completed recruitment.

5. anti-dsDNA is a ok endpoint, but looking at renal disease (GFR, urinary sediment, urine protein is critical to assess how good/effective a therapeutic plan is

We recognise the importance of analysing renal disease, but this study was not limited to patients with renal disease, nor was it solely an efficacy study but was designed as a phase II safety study. All these endpoints will be collected during the trial for all patients irrespective of renal involvement.

VERSION 2 – REVIEW

REVIEWER	Peter H Schur MD Brigham and Women's Hospital Harvard Medical School USA
REVIEW RETURNED	09-Oct-2019
GENERAL COMMENTS	1. Please state that Mepacrine also goes by another name: quinacrine

VERSION 2 – AUTHOR RESPONSE

Reviewer: 3

Reviewer Name: Peter H Schur MD

1. Please state that Mepacrine also goes by another name: quinacrine
We have made this change in the manuscript, page 9.